# LSEA Evaluation of Lipid Mediators of Inflammation in Lung and Cortex of Mice Exposed to Diesel Air Pollution

**DOI:** 10.3390/biomedicines10030712

**Published:** 2022-03-19

**Authors:** Luca Massimino, Alessandra Bulbarelli, Paola Antonia Corsetto, Chiara Milani, Laura Botto, Francesca Farina, Luigi Antonio Lamparelli, Elena Lonati, Federica Ungaro, Krishna Rao Maddipati, Paola Palestini, Angela Maria Rizzo

**Affiliations:** 1Department of Gastroenterology and Digestive Endoscopy, IRCCS Ospedale San Raffaele, 20132 Milan, Italy; admin@lucamassimino.com (L.M.); ungaro.federica@hsr.it (F.U.); 2Molecular Medicine-Neuroscience, Università Vita-Salute San Raffaele, 20132 Milan, Italy; 3School of Medicine and Surgery, University of Milano-Bicocca, 20126 Monza, Italy; alessandra.bulbarelli@unimib.it (A.B.); chiara.milani@unimib.it (C.M.); laura.botto@unimib.it (L.B.); francesca.farina@hotmail.it (F.F.); elena.lonati1@unimib.it (E.L.); 4Polaris Research Centre, University of Milano-Bicocca, 20126 Monza, Italy; 5Department of Pharmacological and Biomolecular Sciences, University of Milano, 20133 Milano, Italy; paola.corsetto@unimi.it; 6Department of Biomedical Sciences, Humanitas University, Pieve Emanuele, 20072 Milan, Italy; luigi.lamparelli@hunimed.eu; 7Department of Pathology, Lipidomics Core Facility, Wayne State University, Detroit, MI 48202, USA; aj2642@wayne.edu

**Keywords:** LSEA, lipid mediators, air pollution, inflammation, diesel exhaust particles

## Abstract

Airborne ultrafine particle (UFP) exposure is a great concern as they have been correlated to increased cardiovascular mortality, neurodegenerative diseases and morbidity in occupational and environmental settings. The ultrafine components of diesel exhaust particles (DEPs) represent about 25% of the emission mass; these particles have a great surface area and consequently high capacity to adsorb toxic molecules, then transported throughout the body. Previous in-vivo studies indicated that DEP exposure increases pro- and antioxidant protein levels and activates inflammatory response both in respiratory and cardiovascular systems. In cells, DEPs can cause additional reactive oxygen species (ROS) production, which attacks surrounding molecules, such as lipids. The cell membrane provides lipid mediators (LMs) that modulate cell-cell communication, inflammation, and resolution processes, suggesting the importance of understanding lipid modifications induced by DEPs. In this study, with a lipidomic approach, we evaluated in the mouse lung and cortex how DEP acute and subacute treatments impact polyunsaturated fatty acid-derived LMs. To analyze the data, we designed an ad hoc bioinformatic pipeline to evaluate the functional enrichment of lipid sets belonging to the specific biological processes (Lipid Set Enrichment Analysis-LSEA). Moreover, the data obtained correlate tissue LMs and proteins associated with inflammatory process (COX-2, MPO), oxidative stress (HO-1, iNOS, and Hsp70), involved in the activation of many xenobiotics as well as PAH metabolism (Cyp1B1), suggesting a crucial role of lipids in the process of DEP-induced tissue damage.

## 1. Introduction

Airborne particulate matter (PM) is a heterogeneous mixture of particles characterized by different sizes, compositions, and sources. Usually, particles are classified into three major size groups: coarse particles (diameter <10 and ≥2.5 μm, PM10), fine particles (diameter <2.5 and ≥0.1 μm, PM2.5), and ultrafine particles (<0.1 μm, UFPs) [1]. Long-term exposure to ambient PM2.5 is considered the fifth-greatest risk factor for global mortality in the Global Burden of Disease, Injuries, and Risk Factors Study 2015; indeed, it determines a high number of premature deaths (i.e., 4–9 million in 2015) [2,3]. The particle size is highly correlated to their toxicity, as a smaller PM size fraction with an increased surface area enhances cell-PM interactions [4,5]. Due to their peculiar features, such as high surface-to-volume ratio, prolonged residence time in the lungs, low clearance efficiency, translocation across air blood air, and transport in lymphatic circulation, UFPs obtained increased attention for their potential toxicity on human health [6]. Of note, is their ability to accumulate in different organs, such as the liver, kidneys, heart, and brain [7,8,9,10,11]. Diesel combustion produces particles of 15–30 nm in diameter contributing mainly to primary UFP emissions [12,13]. Diesel exhaust particles (DEPs) represent the solid fraction of diesel engines [14] and are constituted of a carbon core enriched in high-molecular-weight chemical components, such as PAHs and heavy metals [15,16]. In northern Italy (Lombardy) they represent the most important sources of PM2.5 responsible for the 20% of fine particles (https://www.arpalombardia.it/sites/DocumentCenter/Documents/RAPPORTO%20SULLO%20STATO%20DELL’AMBIENTE%20IN%20LOMBARDIA%20-%202004/08atmosfera.pdf#search=Inventario%20di%20emissioni. Accessed on 9 June 2017). In recent works using an in-vivo model, we demonstrated that DEP exposure led to strong activation of oxidative and inflammatory stress in lung parenchyma [17] and brain [18]. DEPs caused additional ROS production, which affects cell components, such as lipids, finally inducing oxidative damage [19]. Lipid peroxidation is the oxidative degradation of lipids, especially polyunsaturated fatty acids (PUFAs), which generate degradative products, such as epoxides and aldehydes involved in oxidative stress-related pathologies [20,21,22]. Previous work showed that repeated exposure to PM10 in BALB/c mice led to lung lipid reshaping [23]; indeed, we measured an increase of phospholipid and cholesterol content, lipid peroxidation correlated to oxidative stress-induced by PM10. These data suggested possible PM effects on cell membrane structure and functions. The cell membrane provides messenger molecules that mediate cell-cell communication, inflammation, and resolution processes, suggesting the importance of understanding lipid modifications induced by UFPs. In this contest, lipid mediators (LMs) could show opposite or redundant properties. Moreover, the overall balance among various oxygenated PUFA species might greatly impact physiological and pathological conditions, thus challenging a straightforward definition of their ability to modulate biological processes. 

Starting from these considerations, we have investigated DEP outcomes on LM composition in lung and cortex tissues. BALB/c mice were exposed for different times through intratracheal instillation of particulate, a useful and validated in-vivo model to study pollutant-induced acute and subacute toxicity [17,18,24]. The lung and brain were chosen as the first site of DEP accumulation and the target of UFP delivery via the respiratory system, respectively [25]. Indeed, previous evidences indicated that DEPs induce oxidative stress, inflammation, and unfolded protein response in the CNS [17,26,27,28], with the highest status of oxidative stress and inflammation in the cortex [29]. 

Here we introduce an ad hoc bioinformatic pipeline to evaluate lipidomic profile and correlate results with specific biological processes. To this aim, Lipid Set Enrichment Analysis (LSEA) was developed to evaluate the functional enrichment analysis of lipid sets belonging to biological processes found in the following databases: Chemical Entities of Biological Interest (ChEBI), LIPID MAPS^®^ Structure Database (LMSD), Reactome, and Small Molecule Pathway Database (SMPDB). Our results indicate that DEP treatment impacts the tissues in terms of PUFA-derived LM profile in association with specific inflammatory and oxidative stress pathways, with different features related to acute and subacute treatments. Data obtained from the lipidomic analysis highlighted that PUFA-derived LMs correlate with the activation of inflammatory processes, oxidative stress and xenobiotic metabolism.

## 2. Materials and Methods

### 2.1. Animal Housing

Data from the literature indicate that the susceptibility to air pollution is gender-related; as in our previous works, we wanted to analyze the effects of DEP in a homogeneous animal population and for this reason, we used a male mouse model. Before treatments, to acclimate to the housing facility under controlled environmental conditions (temperature 19–21 °C, humidity 40–70%, lights on 7 a.m.–7 p.m.), male BALB/c Ollamhs mice (7–8 weeks old, 20–25 g weight, Envigo) were housed in groups of three in plastic cages for five days; food and water were administered ad libitum. The Institutional Animal Care and Use Committee of the University of Milano-Bicocca approved the protocol and procedures (protocol 02-2014) that complied with guidelines set by the Italian Ministry of Health (DL 26/2014 “Application of the Directive n. 2010/63/EU on the protection of animals used for scientific purposes”).

### 2.2. Intratracheal Instillation

Twelve animals were randomly divided into four experimental groups (n = 3/group) and exposed in the morning to acute and subacute treatments with isotonic solution (CTRL) or Diesel exhaust particles (DEP). The experiments were replicated twice, for a total of 6 CTRL and 6 DEP-treated mice for acute and subacute treatments. The sample size was chosen to minimize the number of animals employed [30]. Every mouse was singularly exposed to a mixture of 2.5% isoflurane (flurane) anesthetic gas and kept under anesthesia during the whole instillation procedure. Once a deep stage of anesthesia was reached, mice were intratracheally instilled using MicroSprayer Aerosolizer system (MicroSprayer Aerosolizer Model IA-1C and FMJ-250 High-Pressure Syringe, Penn Century, USA) with 100 µL of isotonic saline solution (CTRL) or 50 µg of DEPs in 100 µL of isotonic saline solution (DEP) as previously described [17,18]. UFP concentration was chosen on the basis of in-vivo investigations suggesting that a dose of 50 μg/mice is able to induce in lungs acute and sub-acute inflammatory changes [31,32].

Acute and subacute instillation protocols are illustrated in Figure 1 and previously described [33]. The acute treatment consisted of a single instillation of 50 μg DEPs and animal sacrifice after 3 h. Instead, the subacute treatment consisted of three repeated instillations every 3 days and animal sacrifice after 24 h from the last instillation [30,34,35]. Mice of each experimental group were anesthetized by gas to minimize suffering and euthanized with cervical dislocation. As described previously, the animal tissues have been collected [30,35] and evaluated for markers of cytotoxicity, inflammation, and oxidative stress.

### 2.3. UFP Characterization

DEP batches were provided by ENEA in the framework of the project “Biological effects and human health impacts of ultrafine particle sources” led by Prof Camatini of POLARIS research center. Particle sampling procedures and their characterization are extensively elsewhere reported [15]. Briefly, DEPs were sampled from a diesel Euro 4 light-duty vehicle without an anti-particulate filter fueled by commercial diesel and run over a chassis dyno. Aggregates of round carbonaceous particles lower than 50 nm were evident in TEM and SEM images [15]. PAHs and transition metal (Fe, Zn, Cr, Pb, V, and Ni) concentrations were high in DEP. Detailed composition summarized in Appendix A was analyzed by Longhin and colleagues [15].

### 2.4. Lung and Cortex Protein Analysis

The lung and cortex of CTRL and DEP-treated mice were homogenized and assayed for protein content as previously described [30]. The protein amount was determined by the BCA assay (Sigma Aldrich, St. Louis, MO, USA).

Protein characterization was performed by Western blot and revealed by immunoblotting with specific antibodies: rabbit polyclonal heme oxygenase-1 (HO-1) (sc-10789; Santa Cruz Biotechnology™, Dallas, TX, USA), rabbit polyclonal inducible nitric oxide synthase (iNOS) (sc-8310; Santa Cruz Biotechnology™, Dallas, TX, USA), rabbit polyclonal cytochrome 1b1 (Cyp1b1) (sc-32882; Santa Cruz Biotechnology™, Dallas, TX, USA), goat polyclonal heat shock protein 70 (Hsp70) (sc-10-70; Santa Cruz Biotechnology™, Dallas, TX, USA), rabbit polyclonal cyclooxygenase 2 (COX2) (4842; Cell Signalling Technology^®^, Danvers, MA, USA). The secondary antibodies were appropriate horseradish peroxidase (HRP)-conjugated goat anti-rabbit (131460 Thermofisher Scientific™, Waltham, MA, USA) or donkey anti-goat (sc-2020; Santa Cruz Biotechnology™, Dallas, TX, USA). Immunoblot bands have been analyzed as previously described [30]. 

### 2.5. Sample Preparation and LC-MS Analysis of Lipid Mediators

For LC-MS analyses, lung and cortex tissues were homogenized using zirconium beads in a high-frequency oscillator (Precellys, Bertin Technologies, Montigny-le-Bretonneux, France) in phosphate-buffered saline (50 mM phosphate, pH 7.2, and 0.9% sodium chloride). Protein content was quantified using the BCA method. Recovery and quantitation homogenates containing 1–2 mg protein were spiked with 5 ng each of 15(S)-HETE-d8,14(15)-EpETrE-d8, Resolvin D2-d5, Leukotriene B4-d4, and Prostaglandin E1-d4 as internal standards. The samples were then purified using C18 extraction columns as described earlier [36,37,38]. Briefly, the internal standard spiked samples were applied to conditioned C18 cartridges, washed with 15% methanol in water followed by hexane, and dried under vacuum. The cartridges were eluted with 0.5 mL methanol. The eluate was dried under a gentle stream of nitrogen. The residue was redissolved in 50 µL methanol-25 mM aqueous ammonium acetate (1:1) and subjected to LC-MS analysis.

HPLC separation (Prominence XR system, Shimadzu) was achieved using a Luna C18 (3µ, 2.1 × 150 mm) column; the mobile phase consisted of a gradient between A: methanol-water-acetonitrile (10:85:5 *v/v*) and B: methanol-water-acetonitrile (90:5:5 *v/v*), both containing 0.1% ammonium acetate, the flow rate was 0.2 mL/min. The gradient program concerning the composition of B was as follows: 0–1 min, 50%; 1–8 min, 50–80%; 8–15 min, 80–95%; and 15–17 min, 95%. The eluate was directly introduced to ESI source of QTRAP5500 mass analyzer (AB Sciex, Framingham, MA, USA) in the negative ion mode with the following conditions: Curtain gas: 35 psi, GS1 & GS2: 35 psi, Temperature: 600 °C, Ion Spray Voltage: −1500 V, Collision gas: low, Declustering Potential: −90 V. The eluate was monitored by the Multiple Reaction Monitoring method to detect unique molecular ion—daughter ion combinations for each of the lipid mediators using a scheduled MRM around the expected retention time for each compound. Optimized Collisional Energies (18–35 eV) and Collision Cell Exit Potentials (7–10 V) were used for each MRM transition. Spectra of each peak detected in the scheduled MRM were recorded using Enhanced Product Ion scan to confirm the structural identity. T Analyst 1.6.3 software and the MRM transition chromatograms were utilized to collect the data that were quantified by MultiQuant software (both from AB Sciex). The internal standard signals in each chromatogram were used for normalization, recovery, as well as relative quantitation of each analyte [36,37,38].

### 2.6. Statistical Analyses

Differential abundance analysis was performed with DESeq2 [39] based on the normalized signals, namely the counts divided by sample-specific size factors determined by the median ratio of counts relative to geometric mean per lipid species. Low-dimensional embedding of high-dimensional data was achieved by the t-Distributed Stochastic Neighbor Embedding (t-SNE) machine learning algorithm with Rtsne [40] using the variance stabilizing transformed signals. Linear correlation between lipid molecules and protein expression levels was performed with Hmisc [41] using Pearson’s r or Spearman’s rho rank correlation coefficients for parametric and non-parametric distributions, respectively.

### 2.7. Lipid Set Enrichment Analysis (LSEA)

Lipid-biological process associations were downloaded from Chemical Entities of Biological Interest (ChEBI, https://www.ebi.ac.uk/chebi/), LIPID MAPS^®^ Structure Database (LMSD, https://www.lipidmaps.org/), Reactome (https://reactome.org/), and Small Molecule Pathway Database (SMPDB, https://smpdb.ca/) databases, last accessed on August 2021. The Lipid Set Enrichment Analysis (LSEA) pipeline (Massimino, L., Lamparelli, L.A., Rizzo, A.M. and Ungaro, F. Annotation and functional investigation of lipid-related molecular pathways by Lipid Set Enrichment Analysis, LSEA) calculates the absolute and relative abundances of lipid molecules and performs functional enrichment of lipid sets belonging to biological processes using the Gene Set Enrichment Analysis (GSEA) software [42] or similar alternatives, such as GSEApy [43].

## 3. Results

### 3.1. Lung and Cortex LM Signature

PUFAs are known to regulate the duration and magnitude of inflammation, with omega-6 and omega-3 fatty acids acting mostly as inflammatory or pro-resolving molecules, respectively.

To profile their fatty acyl signature, control (CTRL) and DEP treated mouse lung and cortex tissues were analyzed by liquid chromatography-tandem mass spectrometry (LC-MS/MS). Lipid mediators were classified based on their precursors: the omega-6 arachidonic acid (AA), dihomo-linolenic acid (DGLA), linoleic acid (LA), and omega-3 docosahexaenoic acid (DHA), eicosapentaenoic acid (EPA), alpha-linolenic acid (LNA). Moreover, a classification related to the major enzymes involved in their biosynthesis was utilized to better stratify the data.

One hundred and forty LMs were initially identified; out of these, 99 LMs were found in the majority of the samples and were included in our analyses (Figure 2; Appendix A). Two-dimensional embedding of multidimensional LC-MS/MS data by t-distributed stochastic neighbor embedding (t-SNE) machine learning algorithm (Figure 3A) highlighted a distinct lipid signature for the lung and cortex, irrespective of the treatment; 52 LMs were differentially enriched between lung and cortex (Figure 3B,C; Appendix A).

Lipid Set Enrichment Analysis (LSEA), a computational pipeline that calculates differential abundance of single molecules and functional enrichment of lipid-related biological pathways (Massimino, L., Lamparelli, L.A., Rizzo, A.M. and Ungaro, F. Annotation and functional investigation of lipid-related molecular pathways by Lipid Set Enrichment Analysis, LSEA) was used. Functional enrichment analysis highlighted tissue-specificity for pro-resolving lipid mediator biosynthesis pathways, responsible for docosanoids and omega-3 derivatives enrichment in the cortex and omega-6-derived eicosanoids in the lung (Figure 3E; Appendix A).

To assess the possible lipid changes induced by mouse manipulation, mice from CTRL (saline), acute, and subacute groups were compared by LSEA. The analysis indicated that mouse manipulation and saline treatment with subacute protocol upregulated the synthesis of eicosanoids, such as the oxylipins EpDPE and EpETrE in the cortex (Figure 4A–C; Appendix A), two epoxy derivatives with possible anti-inflammatory effects [44]. In the lung (Figure 4D–F), subacute CTRL samples showed significant enrichment in Resolvins in comparison to the acute saline-treated tissues.

### 3.2. Effect of DEP Acute and Subacute Treatment on LMs in Lung and Cortex

To analyze the effect of DEPs on tissue LMs after acute treatment, lung- and cortex-treated tissues were compared with respective CTRL tissues. LSEA showed the acute DEP treatment to downregulate eight specific LMs in the lung, mainly hydroxylated derivatives synthesized by CYP450 enzymes, while no specific metabolic pathways were found to be dysregulated in the lung after acute DEP treatment (Figure 5A,B; Appendix A). By contrast, acute treatment did not induce a specific change in LM concentration in the cortex, although we observed an overall significant increase of eicosanoids, including prostaglandins and thromboxanes (Figure 5C–E; Appendix A).

Subacute exposure to DEPs induced lower effects in all analyzed tissues. In particular, no specific effect was observed in the cortex when compared to saline control, while two LMs, Maresin-1 and 9(10)-EpOME, were found upregulated by DEPs in the lung during subacute treatment (Figure 6A,B; Appendix A). In addition, LSEA showed downregulation of the Resolvin pathways upon DEP treatment in the lung (Figure 6; Appendix A).

Finally, we aimed to assess the differences between the acute and subacute effects of DEPs. The major differences were found in the lung (Figure 7A–D; Appendix A) with the downregulation of many pathways after subacute DEP exposure, in particular omega-3 oxo derivatives and pro-resolving lipid mediators, while, in the cortex, a downregulation of docosanoids and DPA-derived LMs was determined in the subacute protocol (Figure 7E–H; Appendix A).

### 3.3. Inflammatory Protein Markers

Proteins related to oxidative stress and inflammation were previously examined in the lung, heart, hippocampus, cerebellum, and cortex [17]. As shown in the previous study, after DEP exposure in both respiratory and cardiovascular systems, the inflammatory response (COX-2 and MPO), as well as pro- and anti-oxidant proteins (HO-1, iNOS, Cyp1b1, Hsp70), were increased by DEP exposure although the stress persisted only in cardiac tissue under repeated instillations.

As shown in Figure 8, acute and subacute DEP treatments induced upregulation of inflammatory proteins, such as Hsp70, iNOS, HO-1, Cyp1b1, and COX2, particularly evident in the cortex.

To investigate whether the upregulation of these inflammatory protein markers was linked to the observed dysregulation of LMs, we performed a multivariate analysis between protein levels and LM signals and found many LMs were correlated with proteins in both treatment protocols (acute and subacute; Figure 9; Appendix A). In the cortex, after acute treatment, we observed six LMs being positively correlated with COX2, Cyp1b1, and Hsp70, while in subacute treatment all proteins were found to correlate with one or more LMs. Specifically, the negative correlation with HO-1 and the positive with iNOS and Cyp1b1 were found. Cyp1b1 level was also positively correlated with many LMs in the lung during acute treatment, while we found a negative correlation of COX2 and Hsp70 and a positive correlation with HO-1 during the subacute treatment of mice.

## 4. Discussion

Air pollution, in particular UFPs, is one of the most widespread and dangerous environmental toxicants in the world. The World Health Organization estimates that 9 out of 10 people are exposed to polluted air, resulting in 7 million estimated deaths (https://www.who.int/china/home/02-05-2018-who-issues-latest-global-air-quality-report-some-progress-but-more-attention-needed-to-avoid-dangerously-high-levels-of-air-pollution, last accessed on 30 June 2021), caused by respiratory complications, such as pneumonia or chronic obstructive pulmonary disease, and stroke [45].

Moreover, many epidemiological studies support that exposure to air pollution, particularly NO_2_ and PM2.5, increases COVID-19 susceptibility to infection and mortality. The available data suggest that air pollution exposure is correlated to adverse effects and poor prognosis of patients affected by SARS-CoV-2 disease [46].

UFP adverse biological effects are related to (i) their ability to inhibit phagocytosis, and therefore, to enhance interaction with the alveolar epithelium [47]; (ii) their capability to effectively translocate from the respiratory tract to extrapulmonary sites [48,49,50]. Finally, the UFP high surface area-to-mass ratio can adsorb potentially toxic chemicals or metals acting as a source of ROS by increasing proportionally more chemical redox cycling than PM2.5 [51].

DEP exposure, a primary contributor of UFPs, leads to pulmonary inflammation disturbing the alveolar cell differentiation. Indeed, DEPs induce a massive synthesis and secretion of the inflammatory cytokine interleukin-8 by human monocytes [52] and in the respiratory system [53]. Moreover, a response to exposure to high DEP concentration increased C-reactive protein serum content, the classical acute-phase protein [54,55].

In addition, DEPs induce the release of the inflammatory cytokine TNF-α, controlled by the NF-κB expression, activating alveolar macrophages. TNF-α determines, through the upregulation of NF-κB, macrophage apoptosis. Indeed, DEPs stimulate not only the TNF-α gene expression but also the apoptotic responses in alveolar macrophages and consequently, lung inflammation and injury [56]. The process causes respiratory disturbances which lead to superoxide and H_2_O_2_ formation. These ROS stimulate the MEKK-1 (mitogen-activated protein kinases/extracellular-regulated kinase kinase-1), which activates IKK (inhibitor of nuclear factor kappa B (IκB) kinase) and JNK (c-Jun N terminal kinases); both kinases stimulate NF-κB downstream. TNF-α stimulation also causes mitochondrial oxidative stress, calcium release, and finally the phospholipase A2, lipoxygenase, and acid sphingomyelinase activation. All these enzymes are involved in lipid mediator synthesis.

Li et al. demonstrated that in the intestines UFP exposure enhances the concentration of oxidative lipid metabolites, such as AA, HETEs, HODEs, PGD2, and LPA. This evidence suggests an interplay among air pollution, inflammatory responses, and lipid mediators [57].

The fatty acid-derived lipid mediators, produced during different phases of the inflammatory process, are crucial players in the acute inflammatory response [58]. They are synthesized from the omega-6 and omega-3 PUFAs, such as arachidonic, eicosapentaenoic, and docosahexaenoic acids. Indeed, LMs are rapidly synthesized by the innate immune system cells that are recruited to the site of the event [59]. Then, PUFAs are oxygenated through enzymatic or free radical-mediated autoxidation reactions into a great number of bioactive oxygenated LMs [60]. Hundreds of oxygenated species, such as prostanoids and isoprostanes (isoP), leukotrienes, regio- and stereoisomers of mono- and poly- hydroxyl-, hydroperoxy-, epoxy-, and keto-fatty acids are generated by the oxygenation of PUFAs via an enzymatic or non-enzymatic pathways. Cyclooxygenases (COXs), Lipoxygenases (LOXs), and CYP450 are the main classes of enzymes involved in LM synthesis [61].

In the initial inflammatory phase, COX enzymes drive the synthesis of prostaglandins and thromboxanes while LOX enzymes drive the synthesis of leukotrienes and lipoxins [61]. These LMs have different effects on cells by activating GPCRs (G-protein coupled receptors) [62]. COX-and LOX-dependent pathways are clinically targeted in the treatment of inflammation, cardiovascular disease, asthma, fever, and pain [63].

The PUFA hydroxylation and epoxidation are catalyzed by CYP and produce specific sets of LMs, such as the AA-derived epoxy-eicosatrienoic acids (EETs) [61]. By contrast, some CYPs also possess a hydroxylase activity and produce hydroxylated products, such as AA-derived 20-hydroxyeicosatetraenoic acid (20-HETE). These are hormones, growth factors and secondary messengers; they exert opposite roles in the regulation of vascular, renal, and cardiac functions [64].

The early stage of inflammation is crucial for survival, nevertheless, its self-limitation is equally important. Indeed, the failure of the resolution of inflammation might determine chronic inflammatory diseases including cardiovascular and neurological disorders, auto-immune diseases, diabetes, and cancer [65].

The resolution of inflammation is a physiological process carried out by a distinct class of LMs, the specialized pro-resolving mediators, actively orchestrating the return of the tissue to its homeostasis after an acute inflammatory response [66]. These mediators are mainly represented by resolvins (Rvs), lipoxins (LXs), protectins (PDs), and maresins (Masr), synthesized from the omega-3 PUFAs EPA and DHA through many intermediates (18-HpETE, 17-HpDHA, and 14-HpDHA) [61].

It was already demonstrated that PMs modulate lipid metabolism in the small intestine, including the production of HETEs and HODEs via local and systemic pathways [67]. Recently, Lin and collaborators showed that the exposure of PM rich in PAHs results in increased levels of oxidative products of PUFAs (hydroxy-eicosatetraenoic, HETEs, and hydroxyl-octadecadienoic, HODEs acids) as well as increased activity of antioxidant enzymes paraoxonase and arylesterase in the human blood [68]. Furthermore, UFPs induce systemic oxidative stress and inflammation [69,70] via the FA oxidative metabolism in the liver and intestines [71].

The development of a specific pipeline provided us the opportunity to evaluate, in a complex lipidomic data set, the differential abundance of single molecules and functional enrichment of lipid-related biological pathways.

This evaluation clearly indicated that lung and cortex tissues have different LM profiles, mainly correlated to their fatty acid abundance, with the prevalence of omega-3-derived docosanoids in the cortex and omega-6-derived eicosanoids in the lung.

The lipid signature of the analyzed tissues accounts for their different sensitivity to DEP treatment in both acute and subacute phases.

The data presented in our study confirm the role of lipid mediators during the inflammatory processes induced by DEP exposure; moreover, our results indicate that DEP acute exposure significantly upregulates eicosanoid metabolism in the cortex (Figure 5C–E) sustaining the generation of systemic inflammation.

The acute treatment has a greater effect on lipidomics, compared to subacute. Indeed, in the cortex, the subacute treatment was not able to induce LM modulation, while in the lung was still evident with an upregulation of Maresin-1 and 9(10)-EpOME. The functional enrichment showed in lung a downregulation of the Resolvin pathways upon subacute DEP treatment. Maresin-1 is generated from DHA in human peripheral blood mononuclear cells; in primary human vascular smooth muscle and endothelial cells, Maresin-1 decreases the proinflammatory TNF-α effects. On the contrary, 4(±)9(10)-EpOME, synthesized from linoleic acid in neutrophils during the oxidative burst, has been isolated from the lungs of hyperoxic rats and humans with acute respiratory distress syndrome [72]. The (±)9(10)-EpOME is cytotoxic and induces mitochondrial dysfunction, which may be due to the diol metabolites as well as the parent epoxide [73]. Taken together, these data might indicate the presence of chronic inflammation in the lung after DEP subacute treatment, with one pro-resolving mediator that is up-regulated, while the general pathway of resolvins is downregulated and the EpOME indicates the persistence of oxidative stress.

In our study, LMs also correlate with protein markers of inflammation and oxidative stress that are greatly upregulated by acute and subacute DEP treatment in cortex tissues. Interestingly, even if LMs are not significantly modified after subacute treatment, they strongly negatively correlate with HO-1 and positively with COX2 and iNOS. HO-1 is a crucial enzyme for the antioxidant response and neuroprotection and the negative correlation with 17-HDoHE and 13-OxoODE, DHA-derived markers of oxidative stress, sustains its role as a ROS protective enzyme in the brain.

Many other correlations might be discussed within our data and are the results of the fine statistical evaluation of the protein and lipidomic analyses.

A limitation of this kind of study was revealed from our comparison between acute and subacute CTRL mice, which indicates the animal model sensitivity to manipulation, with significant LM variation induced by the procedure of saline instillation; this data has to be taken into serious consideration when working with omic sciences.

In conclusion, our result pointed out the impact of DEP exposure on lipid mediator metabolism with specific and different pathways in lung and brain tissues, correlating also with the protein mediating inflammatory state. In this lipidomic analysis, we set up a bioinformatic approach, the Lipid Set Enrichment Analysis, which allows deep omic data analysis to obtain insight into the functional metabolic pathways related to their variations.

## Figures and Tables

**Figure 1 biomedicines-10-00712-f001:**
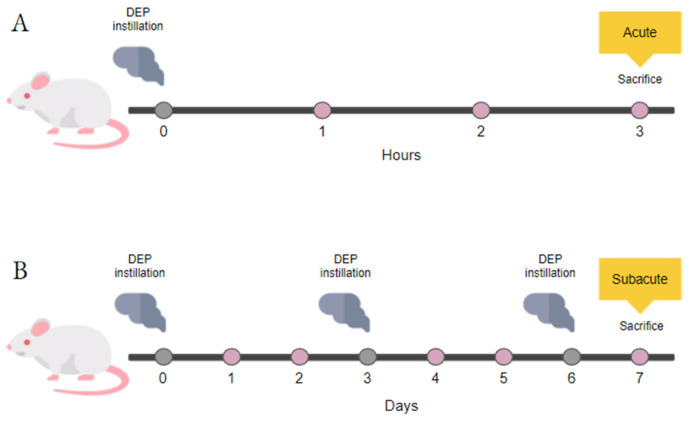
Study design. Schematic representation of BALB/c acute (**A**) and subacute (**B**) treatments.

**Figure 2 biomedicines-10-00712-f002:**
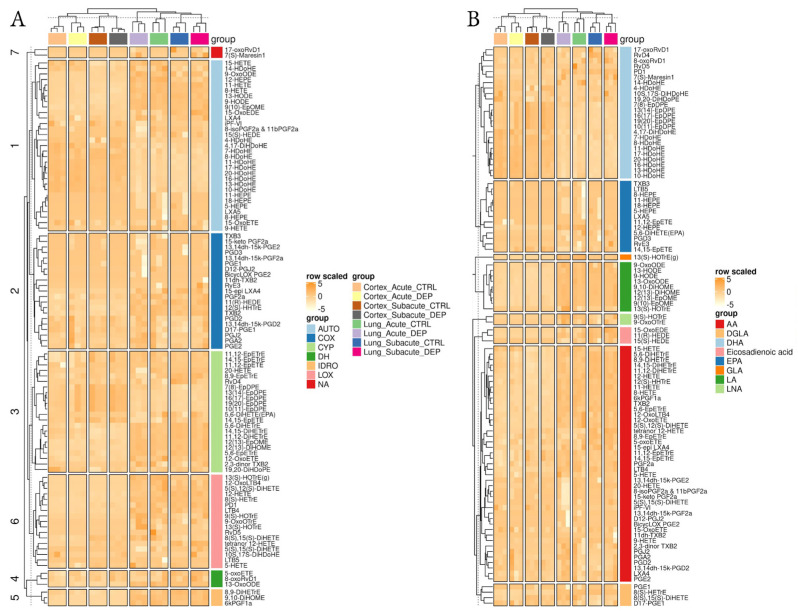
Overview of lipidomic results. Heatmaps of normalized lipid mediator signals among the different conditions and tissues, clustered by the main synthetic enzyme (**A**) or their fatty acid precursor (**B**).

**Figure 3 biomedicines-10-00712-f003:**
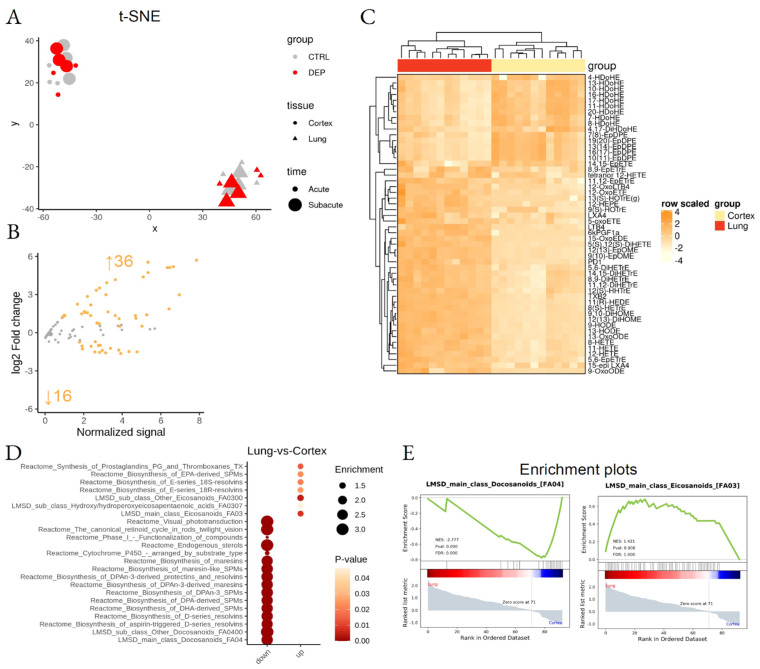
Differential lipid mediator enrichment between lung and cortex, upon acute and subacute DEP treatments. (**A**) Sample dispersion within the t-SNE multidimensional scaling space, where the group, tissue, and treatment time are coded by color, shape, and size, respectively. (**B**) MA plot showing normalized lipid mediator abundance as a function of log2 fold change. Orange dots and text identify the statistically significant differentially abundant lipid species. (**C**) Heatmap of the differentially abundant lipid species. (**D**) LSEA balloon plot showing the differentially functionally enriched molecular pathways. Size and color identify dataset enrichment scores and statistical significance, respectively. (**E**) LSEA enrichment plots of two examples of differentially functionally enriched molecular pathways, showing the enrichment score as a function of the dataset rank between the two conditions. Up and down arrows show the number of differentially abundant lipid mediators.

**Figure 4 biomedicines-10-00712-f004:**
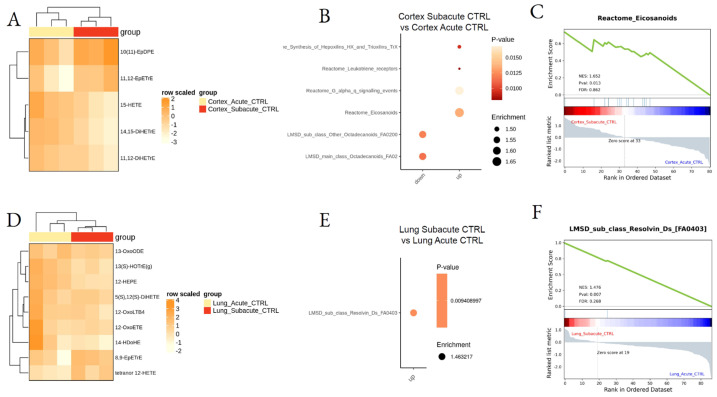
Differential lipid mediator enrichment between acute and subacute control treatments, in lung and cortex. Heatmap of the differentially abundant lipid species (**A**,**D**), LSEA balloon plot showing the differentially functionally enriched molecular pathways (**B**,**E**), and LSEA enrichment plot examples (**C**,**F**), between Acute and Subacute DEP treatments in cortex and lung.

**Figure 5 biomedicines-10-00712-f005:**
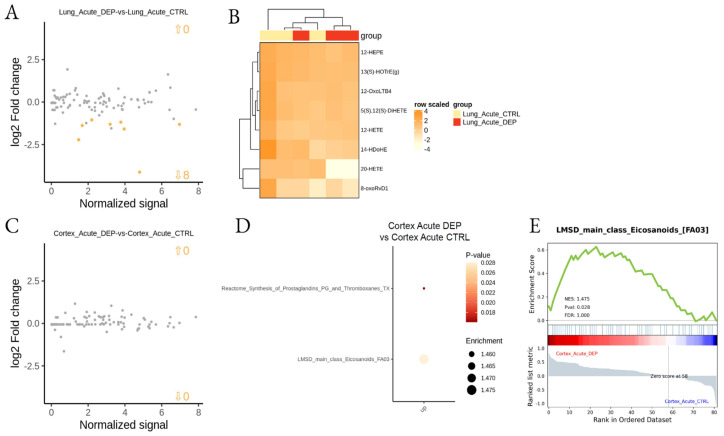
Differential lipid mediator enrichment between acute DEP and control treatments, in lung and cortex. MA plot showing normalized lipid mediator intensity as a function of log2 fold change (**A**) and heatmap of the differentially abundant lipid species (**B**), between DEP and control acute treatments, in the lung. MA plot showing normalized lipid mediator intensity as a function of log2 fold change (**C**), LSEA balloon plot showing the differentially functionally enriched molecular pathways (**D**), and an LSEA enrichment plot example (**E**), between DEP and control acute treatments, in the cortex. Up and down arrows show the number of differentially abundant lipid mediatiors.

**Figure 6 biomedicines-10-00712-f006:**
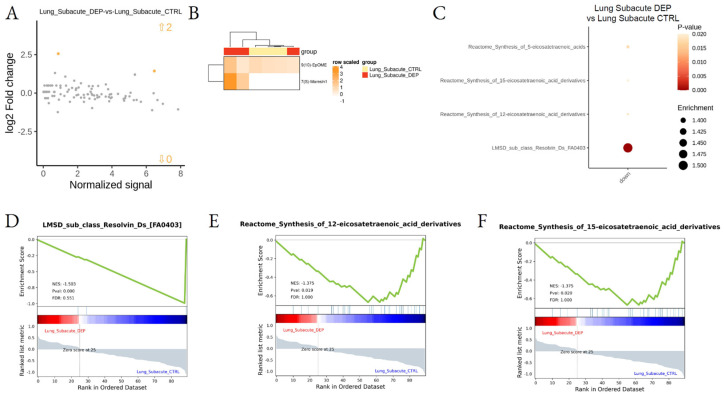
Differential lipid mediator enrichment between subacute DEP and control treatments, in Lung. MA plot showing normalized lipid mediator intensity as a function of log2 fold change (**A**), heatmap of the differentially abundant lipid species (**B**), LSEA balloon plot showing the differentially functionally enriched molecular pathways (**C**), and three LSEA enrichment plot examples (**D**–**F**), between subacute DEP and control treatments, in the lung. Up and down arrows in A show the number of differentially abundant lipid mediatiors.

**Figure 7 biomedicines-10-00712-f007:**
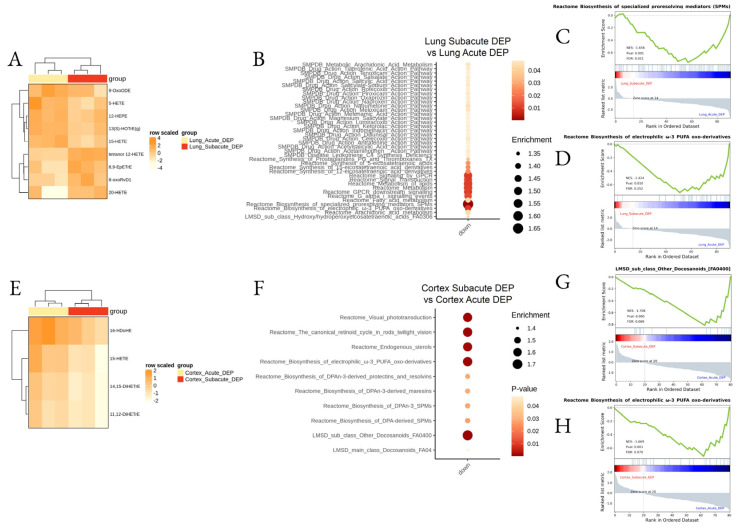
Differential lipid mediator enrichment between acute and subacute DEP treatments, in the lung and cortex. Heatmap of the differentially abundant lipid species (**A**,**E**), LSEA balloon plot showing the differentially functionally enriched molecular pathways (**B**,**F**), and three LSEA enrichment plot examples (**C**,**D**,**G**,**H**), between acute and subacute DEP treatments, in the lung and cortex.

**Figure 8 biomedicines-10-00712-f008:**
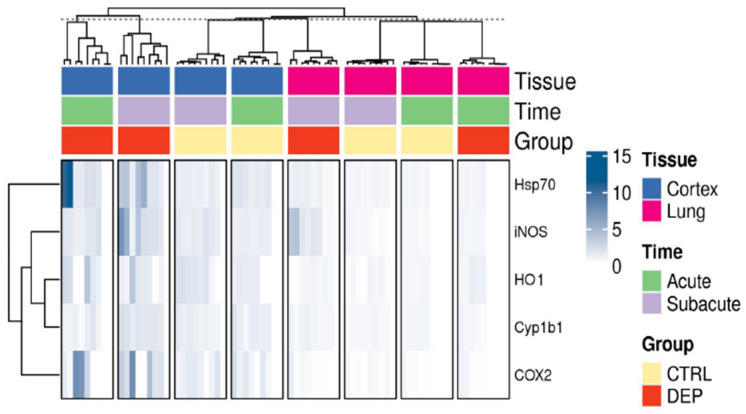
Overview of inflammatory protein markers. Heatmap of the relative western blot signals among the different conditions and tissues.

**Figure 9 biomedicines-10-00712-f009:**
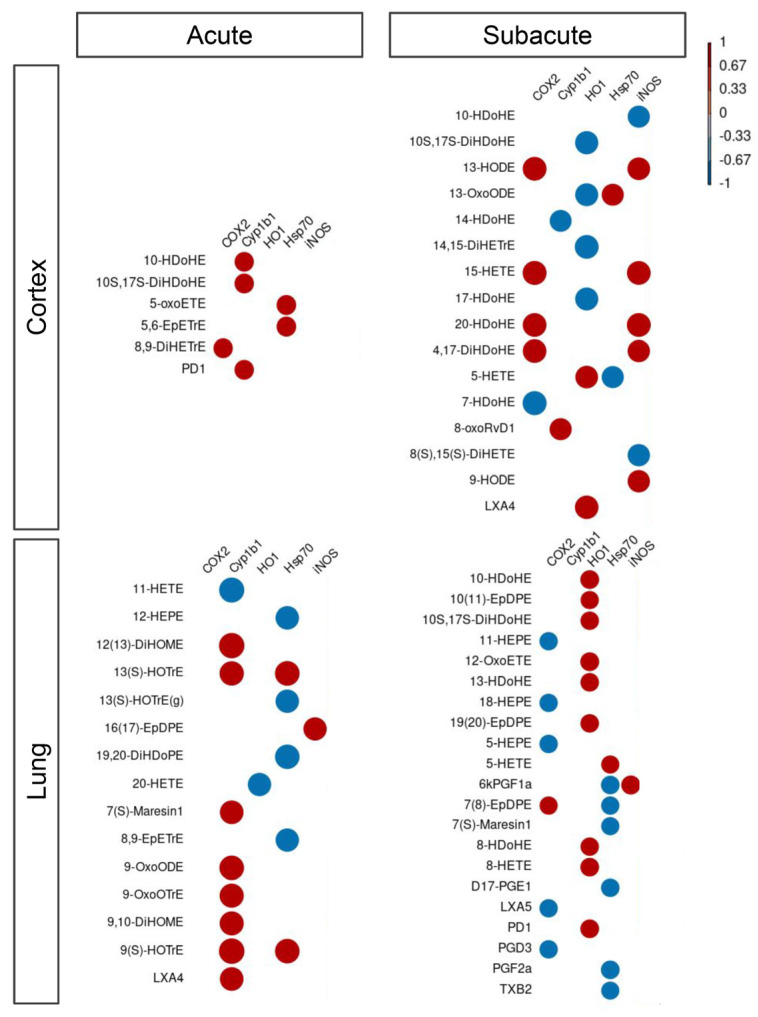
Correlation between lipid mediators and inflammatory protein markers. Correlation between lipid mediators and inflammatory protein markers within the different conditions, namely Cortex and Lung upon acute or subacute DEP treatments. Color scaling shows positive and negative correlations in red and blue, respectively.

## Data Availability

All the data supporting reported results can be found in Appendix A.

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
