# Peer review of "LSEA Evaluation of Lipid Mediators of Inflammation in Lung and Cortex of Mice Exposed to Diesel Air Pollution"

_biomedicines, 2022, doi:10.3390/biomedicines10030712_

Round 1
Reviewer 1 Report
Dear the Editor,
Massimino L et al described an alteration of lipid mediators determined by lipidomics approach in murine model of DEP exposure. They found an elevation of Resolvin and Maresin, indicative of the end of inflammatory process. This manuscript referred to the property and pathophysiological role of particulate materials. In sharp contrast, the presentation of data appeared to be rather poor with no clear conclusion of this experiment.
Major issue to be considered:
1) Please prepare figures and letters appeared there more readably.
2) Please describe the conclusion more clearly. In result section, please indicate of which data supports this conclusion.
3) Method section needs to be appeared in the main text.
4) Introduction and Discussion seem lengthy. Please describe topics only related to the result.
Author Response
Please find attached file with reviewer response

Reviewer 2 Report
The authors of the manuscript entitled “LESA evaluation of lipid mediators of inflammation in lung and cortex of mice exposed to diesel air pollution” studied the effect of DEP on LMs in lungs and cortex of mice and designed a bioinformatic pipeline LESA to aid data analysis. The overall design of the experiment was logical and sound. My minor comments are:
The authors provided sufficient background knowledge in the introduction, which is very helpful for the audience. However, the structure of the introduction needs some improvement to make it more logical and organized. One example is that the last three paragraphs could be integrated into one.
In this experiment, only male mice were used (Line 110). Explanation of any specific reasons why female mice were not included is needed.
The description of intratracheal instillation experiment groups is not clear (Line 120-144). The authors mentioned that there were two experimental groups. According to the information provided, there were four groups included in the experiment. It would be helpful if the authors can provide a table with groups or make the information clearer in their Figure 1.
In the statistical analyses part, the three paragraphs could be merged into one.
Figures in this manuscript appear fussy, images with higher resolution are needed.
In the discussion, the authors provided sufficient information about related findings in previous works. However, the structure of the discussion can be improved. For example, the authors could compare their findings with previous findings and discuss the similarity and differences, and the potential reasons of the differences.
The English language of this manuscript needs improvement to make sure international audience understand it. Some examples are lines 42, 44, 78, 94, 138, 150, 159, 271.
The abbreviation of PUFA is lacking in the abstract. In Figure 1, the abbreviation of FMT is lacking.
Author Response
Please find attached file with all 3 reviewers comments

Reviewer 3 Report
Dear Authors,
I found your manuscript very interesting and well-written.
The Introduction provides sufficient background information, Methods are adequately presented, Results are comprehensive and the Discussion extensive.
However, I must urge you to double-check phrase constructing, grammar, abbreviations and format, as typos are evident in the manuscript. Figures are sort of blurry – better resolutions would be an improvement.
More importantly, please make sure that all references are properly introduced in the manuscript – web page can be formatted in the proper reference typeset.
The Conclusions’ phrase should be rephrased as to better encompass your findings and give the reader an overview of your paper. Also, maybe a sentence with the study’s limitations would be appropriate.
Author Response

(The authors gave the same response as above.)

Round 2
Reviewer 1 Report
Dear the Editor
The revised manuscript corrected my concerns properly.
Author Response
We thank the reviewer for her/his suggestion that improved our paper.